# Atopic Dermatitis and Food Allergy: A Complex Interplay What We Know and What We Would Like to Learn

**DOI:** 10.3390/jcm11144232

**Published:** 2022-07-21

**Authors:** Niki Papapostolou, Paraskevi Xepapadaki, Stamatis Gregoriou, Michael Makris

**Affiliations:** 1Allergy Unit, 2nd Department of Dermatology and Venereology, Medical School, National and Kapodistrian University of Athens, Attikon University Hospital, 12462 Athens, Greece; 2Allergy Department, 2nd Pediatric Clinic, National and Kapodistrian University of Athens, 11527 Athens, Greece; vickyxepapadaki@gmail.com; 3Department of Dermatology-Venereology, Faculty of Medicine, National and Kapodistrian University of Athens, Andreas Sygros Hospital, 16121 Athens, Greece; stamgreg@yahoo.gr

**Keywords:** atopic dermatitis, food allergy, food sensitization, prevention strategies

## Abstract

Atopic dermatitis (AD) is a chronic inflammatory skin disorder characterized by intense pruritus, eczematous lesions, and relapsing course. It presents with great clinical heterogeneity, while underlying pathogenetic mechanisms involve a complex interplay between a dysfunctional skin barrier, immune dysregulation, microbiome dysbiosis, genetic and environmental factors. All these interactions are shaping the landscape of AD endotypes and phenotypes. In the “era of allergy epidemic”, the role of food allergy (FA) in the prevention and management of AD is a recently explored “era”. Increasing evidence supports that AD predisposes to FA and not vice versa, while food allergens are presumed as one of the triggers of AD exacerbations. AD management should focus on skin care combined with topical and/or systemic treatments; however, in the presence of suspected food allergy, a thorough allergy evaluation should be performed. Food-elimination diets in food-allergic cases may have a beneficial effect on AD morbidity; however, prolonged, unnecessary diets are highly discouraged since they can lead to loss of tolerance and potentially increase the risk of IgE-mediated food allergy. Preventive AD strategies with the use of topical emollients and anti-inflammatory agents as well as early introduction of food allergens in high-risk infants seem promising in managing and preventing food allergy in AD patients. The current review aims to overview data on the complex AD/FA relationship and provide the most recent developments on whether food allergy interventions change the AD course and vice versa.

## 1. Introduction

Atopic dermatitis (AD) is a chronic inflammatory skin disorder affecting more than 230 million people worldwide [1,2]. With an increasing global prevalence, up to 20% of children and 10% of adults in high-income countries suffer from AD [3,4,5], while AD’s morbidity burden designates the disease as the skin disorder with the highest impact on quality of life in terms of disability-adjusted years [6].

AD presents with great heterogeneity in terms of severity, clinical features, and course, with a complex underlying pathophysiology [5], which involves an interplay between the dysfunctional skin barrier, immune dysregulation—starting with a core T helper 2 (TH2) response accompanied by IgE sensitization to environmental allergens and progressing with a widening of the adaptive immunity with TH1, TH17, and TH22 responses—and skin microbiome dysbiosis [7,8]. In addition, genetic susceptibility, along with filaggrin gene’s mutations being the central but not the only recognized genetic disorder, and environmental factors such as ultraviolet radiation, air pollution, water hardness, household hygiene, and climate change contribute to the multidimensional model of atopic dermatitis [9].

As allergic diseases have reached epidemic proportions, with the first epidemic wave of respiratory allergy increasing about 50 years ago, we are now riding a second epidemic wave of “food allergy” [10,11]. In respect to the TH2 endotype, atopic dermatitis has been proposed as the first “step” in a long although debatable pathway known as the “atopic march”, with food allergy, allergic rhinitis, and asthma presenting either concomitantly or at a later stage [12].

Food allergy (FA) is defined as a food hypersensitivity reaction mediated by immunologic mechanisms, while the term IgE-mediated food allergy is used when the role of IgE is underlined [13]. Food sensitization refers to the production of food-allergen-specific IgE but is not synonymous with food allergy, as individuals can produce specific IgEs to foods without presenting symptoms upon exposure. Hence, sensitization is prerequisite but not synonymous with food allergy [14]. Food allergen sensitization can be identified by skin prick testing or in vitro immunoassays for specific IgE to whole-allergens extracts or allergens components (pure allergen proteins) [15,16]. Not only IgE-mediated food allergy but also other endotypes of food allergy (either non-IgE-mediated or mixed) have been implicated either in the exacerbations or in the morbidity of AD complexing even more the landscape of atopic dermatitis–food allergy interaction [17,18] (Figure 1). The present review aims to provide data on the complex atopic dermatits–food allergy relationship and recent developments on whether food allergy interventions contribute to atopic dermatitis clinical course and vise versa.

## 2. How Much Atopic Is Atopic Dermatitis?

### 2.1. Defining Atopy in Atopic Dermatitis

The term atopy originates from the Greek world “atopos”, which means “out of place”, and the disease was first described as “atopic dermatitis” in 1923 by Cooke and Coca to indicate the concomitant occurrence of IgE-mediated hypersensitivity reactions [8,19]. Atopy is defined as the tendency, whether personal or familiar, to produce IgE antibodies in response to commonly encountered environmental allergens, potentially followed by the development of asthma, rhinoconjuctivitis, eczema/atopic dermatitis, and/or food allergy [13]. Nevertheless, not all patients with atopic dermatitis present IgE allergic sensitization, and thus, they are not atopic [5]. It is well-established that food allergy does not always accompany AD. Almost 50% of children and 35% of adults with AD are sensitised to common environmental and food allergens, with a prevalence ranging between 7–80% among different study populations [20,21]. In regard to food allergens, sensitization rates in AD patients vary from 30 to 80%, but the actual clinically relevant food allergy proportions may be lower especially in less-severe phenotypes of AD [21,22]. In fact, 20–30% of patients with AD, particularly those with mild disease, do not have evidence of IgE sensitization to food allergens or symptoms suggestive of food allergy [9,20]. Food allergy prevalence as a concomitant to AD disease has been reported at significantly higher rates compared to the general population, while respective rates are noted in up to 80% of patients with early-onset, severe, or persistent AD [23].

### 2.2. Phenotypes and Endotypes of AD

#### 2.2.1. Phenotypes of AD

Phenotypes represent categorization according to the clinical characteristics of the patients, such as the presence of IgE sensitization, age of disease onset, presence of acute or chronic lesions, age-related distribution, severity of the disease, or ethnic origin of the patients [24]. Furthermore, AD can be classified according to IgE levels into extrinsic or allergic and intrinsic or non-allergic. The classic extrinsic phenotype (60–80%) is characterised by increased levels of serum IgE, eosinophils, filaggrin mutations, and either personal and/or family atopic background and/or concurrent atopic diseases. The less-common intrinsic phenotype (20–40%) is characterised by normal IgE levels, absence of IgE sensitizations and other atopic manifestations, delayed disease onset, and less transepidermal water loss (TEWL) [25,26]. The European Task Force on Atopic Dermatitis (ETFAD), in order to keep both IgE-associated extrinsic and non-IgE-associated intrinsic endotypes of AD, suggests a slightly different definition of atopy: the familial tendency to develop TH2 responses against environmental antigens [27]. Recently, Nordic AD experts classified adult moderate-to-severe AD into six distinct patient profiles reflecting the spectrum of AD adult phenotypes. In this six-cluster phenotypic classification, cluster 2 included moderate to severe AD patients with type I allergies with increased prevalence of food allergies from childhood, allergic asthma, rhinoconjuctivitis, and family history of atopic diseases, indicating that specific AD phenotypes are highly associated with food allergy [28].

#### 2.2.2. Endotypes of AD

Although AD is considered a predominantly type 2 (T2)-driven disease, it has now become clear that involvement of other T cells, such as TH1, TH17, and TH22, contribute to AD pathogenesis as well [29]. In a study attempting to identify distinct clusters of moderate-to-severe AD patients based on serum biomarkers such as serum total IgE, allergen specific IgE, and serum mediators (chemokines, epithelial and immunomodulatory cytokines, remodeling biomarkers etc.), Thijs et al. described four different AD clusters representing distinct AD endotypes, thus facilitating a personalized therapeutic approach. The first cluster (Cluster 1) was characterized by high six-area six-sign atopic dermatitis (SASSAD) severity score, and high levels of chemokines, tissue inhibitor of metalloproteinases 1, and CD14. Cluster 2 had low SASSAD score and low levels of INF-a, VEGF, and tissue inhibitor of metalloproteinases 1, while Cluster 3 had high SASSAD and low INF-b, IL1, and epithelial cytokines. The fourth cluster had low SASSAD scores but the highest levels of inflammatory markers IL1, IL4, IL13, and TSLP, which are potentially linked to allergy-associated diseases [30]. More recently, the same clusters in severe AD patients have been verified in three out of four previously described clusters. Cluster A was characterized as “skin-homing chemokines/IL-1R1–dominant”, whereas cluster B was “TH1/TH2/TH17-dominant” cluster, cluster C “TH2/TH22/PARC-dominant”, and cluster D “TH2/eosinophil-inferior” [29]. Although pediatric and adult AD share common pathogenetic pathways, certain differences have been acknowledged. The morphology and distribution of AD lesions differ between age groups, with face, trunk, and extensor limb being involved in infants and young children, while chronic lichenified, dry lesions at flexural areas are most observed in adults [26,31]. In adolescents, lichenification at flexures gradually appears, indicating chronicity [31]. These dissimilarities may be the outcome of differences in endotype skewing over the years. Czarnowicki et al. reported that a TH2-dominant immunophenotype was present in infants with moderate-to-severe AD, while a merged TH2/TH22 endotype characterizes adults with moderate-to-severe AD [32]. These discrepancies may reflect the progression of acute to chronic state, but the complex skin cytokine profile in infants characterized by TH1/TH2/TH17/TH22/TH9 cytokines proposes more complex than chronicity mechanisms [33]. In accordance, in a recent exclusively pediatric cohort of 240 patients with AD severity ranging from mild to severe, a cluster analysis based on serum biomarkers identified four distinct clusters. When those clusters were compared to those previously described above in adult populations with moderate-to-severe AD, only one of them showed similar features: the TH1 cell/TH2 cell/TH17 cell/IL-1 dominant cluster (cluster C). Hence, despite differences regarding AD severity between study populations, potential different underlying immunopathologic mechanisms between adults and children are indicated [34]. To conclude, atopic dermatitis is a predominantly type-2-mediated disease, and although the presence of IgE is common in AD patients, and high IgE levels is one of the major diagnostic criteria of AD, it cannot be characterized as a type I hypersensitivity reaction, and hence, the precise role of IgE in the disease is not yet fully elucidated.

## 3. Association between Food Sensitization and Food Allergy with Atopic Dermatitis

### 3.1. Transepidermal Water Loss and Skin Barrier Impairment

A close relationship between atopic dermatitis, food sensitization, and food allergy, especially in childhood, is well-established. Epidermal barrier dysfunction and transepidermal water loss (TEWL) constitute a cornerstone in atopic dermatitis pathophysiology and precedes both the occurrence of atopic dermatitis and food allergy [21,35,36]. Kelleher et al. found that increased TEWL at two days after birth was associated with increased risk of development of AD at one year of age [35]. A constantly growing body of evidence supports that food allergy follows atopic dermatitis diagnosis and leads to the reasonable conclusion that AD is involved in the causal pathway towards food allergy [21,23,37]. Although less commonly, food allergy might precede AD onset in a small number of children [38]. In the Isle of Wight cohort, filaggrin loss-of-function mutations were associated with food sensitization in the early years and food allergy later in childhood, suggesting that skin barrier function per se is important in the development and persistence of food allergy [39].

### 3.2. Food Sensitization and Food Allergy among Patients with AD

Numerous studies support that more severe phenotypes of AD are associated with more frequent diagnosis of food allergy, ranging between 33% to 39%, with occasional studies reporting higher rates up to 80% [40,41,42,43], while food allergy prevalence in the general population is estimated about 0.1–6% [44]. Hence, atopic dermatitis is proposed as a major risk factor for food sensitization and IgE-mediated food allergy [23,45]. Population-based studies have shown that the likelihood of food sensitization is up to six times higher at 3 months of age in patients with AD compared to healthy controls [23]. When including patients with established AD, the prevalence of food sensitization is up to 66%, while proven food allergy by oral food challenges is up to 81% [23]. The Danish Allergy Research cohort (DARC) showed that up to 53% of children with AD, aged 6 months to 6 years, were sensitized to food allergens, while food allergy was confirmed in 15% of them [21]. In the Health Nut study, a large population-based Australian study (*n* = 4453), infants with AD were 6 times more likely to have egg allergy (95% CI 4.6, 7.4) and 11 times more likely to have peanut allergy (95% CI 6.6, 18.6) by 12 months than infants without AD at 12 months of age [46]. Although previous studies suggested that the presence of AD, particularly in its more severe form, is associated with a prolonged course of milk and egg allergy [47,48], recent data were not confirmatory [49]. Such discrepancies might be attributed to different study designs and populations with different disease severity.

### 3.3. Conclusions

Although AD is proposed as a major risk factor for food sensitization and food allergy, its occurrence depends on both severity and chronicity of AD. Rates of food sensitization are high in patients with AD, but frequency of IgE-mediated food allergy confirmed by oral food challenges varies, with more severe AD linked more frequently with food allergy.

## 4. Mechanisms Explaining how Atopic Dermatitis Promotes Sensitization to Food Allergens: The Dual-Allergen-Exposure Hypothesis

### 4.1. Introduction

Sensitization to food allergens can be promoted through in the gastrointestinal tract, oral cavity, skin, and occasionally in the respiratory tract. Recent evidence supports that early allergen exposure through the skin, particularly in respect to peanuts in patients with atopic dermatitis, results in sensitization and subsequent food allergy, whereas early oral exposure promotes tolerance, a theory that is well-known as “the dual-allergen-exposure hypothesis” [38,50].

### 4.2. Allergen Exposure through the Skin

Skin barrier defect is a cardinal characteristic of AD pathogenesis, contributing to skin inflammation and subsequent clinical symptoms of AD. Environmental allergens, including food allergens, can penetrate through the impaired inflamed barrier and potentially promote food sensitization [36,51] (Figure 2). Clinical studies support the hypothesis of epicutaneous sensitization, as it was suggested in the Avon Longitudinal Study of Parents and Children cohort, including 13,971 preschool children, where the application of peanut oil on inflamed skin was significantly associated with peanut allergy [52]. Furthermore, environmental exposure to peanuts and almonds, detected in the house dust samples, has been associated with increased rates of peanut sensitization and allergy in children with AD even without reporting previous consumption [53,54].

Following penetration of food allergens, resident dendritic cells capture the invasive antigens and migrate to draining lymph nodes, where the specific peptide epitopes and occasionally carbohydrate epitopes from allergen molecules are further processed and presented to naïve CD4+ T cells. Inside the lymph nodes with the presence of IL4 and endothelial-derived type 2 cytokines or “alarmins” (mainly IL33 and TSLP), a TH2 polarization occurs, and allergen-specific CD4+ T cells produce large amounts of IL4 and IL13. Subsequently, IgE class switching takes place and drives the production of allergen-specific IgE (sIgEs) from mature plasma cells. The sIgEs bind to high-affinity FcεRI receptors on the surface of mast cells and basophils, leading to sensitization and subsequent food allergy [51]. Innate immunity contributes to T2 inflammation with innate lymphoid cells type 2 (ILC2s) by enhancing the activity of Th2 cells, eosinophils, mast cells, and their cytokines [55]. However, in AD pathogenesis, the presence of IgE is more of a bystander. Type 2 immune responses and inflammation are induced by defects in the stratum corneum (SC), with filaggrin gene mutations being the most common deficiency [56]. The underlying skin layers are exposed and produce alarm signs (IL33 and TSLP), which orchestrate the inflammation. Type 2 cytokines, especially IL-13 and IL-4, enhance the inflammation by opening the tight junction barrier and allowing barrier “leakiness” into the skin [57]. Skin microbiome dysbiosis also plays a role in predisposing skin barrier dysfunction and thus epicutaneous sensitization and clinical food allergy. Up to 90% of AD patients are colonized by *Staphylococcus aureus* (*S. aureus*), and in fact, pathogenic *S. aureus* strains are more frequently present in AD skin than in healthy controls [58]. Moreover, stratum corneum abnormalities, including abundance of *S. aureus* and microbiome dysbiosis, are believed to be able to distinguish children with AD and food allergy from those with AD and no food allergy even in the non-lesional skin [58]. In this study, Leung et al. used a multi-omics approach with skin tape strip-sampling method to show that children with AD and food allergy (AD FA+) represent a unique endotype that distinguishes them from AD without food allergy (AD FA−) and nonatopic controls (NA). A positive correlation between transepidermal water loss and *Staphylococcus aureus* abundance in non-lesional skin of AD FA+ but not AD FA− or NA was revealed. Filaggrin breakdown products in non-lesional skin of AD FA+ children were significantly lower than those on AD FA− and NA, suggesting that the stratum corneum of non-lesional skin in AD FA+ children has unique properties [58]. Furthermore, *S. aureus* superantigens can generate a powerful immune response; non-specific activation of T cells and IgE-mediated downstream inflammatory responses are parts of this response [59,60].

### 4.3. Allergen Exposure through Gastrointestinal Mucosa

Early oral exposure to food allergens leads the antigen-presenting cells in the gut mucosa to recognize the antigens and tο promote naïve CD4+ T cells within the mesenteric lymph nodes to differentiate into α4β7+ gut-homing T cells. These cells induce the production of Foxp3+ Tregs and production of cytokines IL10 and TGF-b that exert immunoregulatory action by promoting the production of IgG4 antibodies and inhibiting Th2-dependent allergenic responses. Hence, an early epicutaneous sensitization due to loss of skin integrity and the concomitant loss of oral tolerance can result in sensitization and clinical food allergy [36,61,62].

### 4.4. Conclusions

The impaired skin barrier in AD is the initial trigger that promotes a variety of pathologic immunological responses to environmental stimuli, including foods, leading to a vicious cycle of atopic disorders; food allergy and atopic dermatitis are probably the most well-correlated disorders especially in early childhood, each one influencing the severity and the natural course of the other (Figure 3).

## 5. Patterns of Clinical Reactivity to Foods in Children with AD

There are three patterns of clinical reactivity to foods in children with AD: (a) immediate-type reactions (IgE-mediated within the first 2 h of consumption), (b) delayed exacerbation of AD (non-IgE-mediated), and (c) mixed reactions with a combination of IgE- and non-IgE-mediated clinical features.

### 5.1. Immediate-Type Reactions (IgE-Mediated)

Immediate-type reactions manifest with either cutaneous symptoms (urticaria, angioedema, flushing) or, in the context of anaphylactic reactions, also including symptoms from the respiratory tract and/or gastrointestinal and/or cardiovascular symptoms (tachycardia, arrhythmias, hypotension, cardiac arrest) [63]. These reactions usually occur within the first 2 h of consumption, ranging from mild single-organ reactions to severe, life-threatening anaphylaxis [63,64].

### 5.2. Delayed Exacerbation of AD (Non-IgE-Mediated)

Delayed, non-IgE-mediated reactions usually occur 6–48 h after consumption of the implicated food allergens, in most cases, as atopic dermatitis flare up [65]. The pattern of delayed exacerbation of AD is not clearly defined in adult-onset AD. Manam et al. reported that challenge-proven food allergy in adults with AD is uncommon despite higher rates of food sensitization compared to healthy controls. Half of patients with confirmed delayed food allergy by food challenge reported significant decrease of AD exacerbations after strict elimination diet of the offending foods [63].

### 5.3. Mixed Reactions

Up to 40% of children experience more complex symptoms combining IgE-mediated symptoms and AD exacerbation [66]. In a study of 64 children with AD who underwent 106 double-blind placebo-controlled food challenges to hen’s egg, milk, wheat, or soy, a mixed reaction was observed in 45%, while 12% had only a delayed reaction [67]. However, in the Danish Allergy Research cohort (DARC) study, 95% of the patients during the double-blind placebo-controlled challenges developed the immediate-type reaction [68]. Conclusively, the frequency and pattern of clinical reactivity varies among patients with AD.

## 6. Diagnosis and Management of Food Sensitization and Food Allergy in Patients with AD

### 6.1. Sensitization to Food Allergens in AD Patients

In clinical practice, the identification of IgE sensitization can sometimes complicate the management of AD. Food sensitization and food allergy are not synonymous, and thus, not all sensitized patients will react upon exposure to certain food allergens [40]. Knowing the high asymptomatic food allergen sensitization rates in patients with AD and that AD is predominantly a TH2-cell-mediated disease, with IgE being more a “bystander” than an actual driver in the disease, management of food sensitization and allergy in AD patients can be challenging.

In case of patients with AD and a compatible history of IgE-mediated food allergy, in vivo and/or in vivo allergy diagnostic procedures should be conducted. According to international guidelines, food-specific skin tests or sIgEs should be considered in all children below 5 years with moderate-to-severe AD, especially those unresponsive to topical treatment [69]. In the light of Learning Early About Peanut Allergy (LEAP) study results, new addendums were made in the National Institutes of Allergy and Infectious Diseases (NIAID) published consensus in regard to peanut allergy: (a) In infants with severe AD and/or egg allergy, measurement of peanut IgE level and/or SPT should be conducted before introduction into the diet; (b) infants with mild-to-moderate AD should not be tested for peanut sensitization, and peanut should be introduced in diet approximately at six months of age; and (c) infants without AD should be free in introduction of peanut and other foods according to the wish of the family [69,70].

### 6.2. The Role of Skin Prick Tests, Atopy Patch Tests, In Vitro Measurement of Specific IgE, and Measurement of Food Allergens Components in the Diagnosis of Food Allergy

The negative predictive value of SPTs to foods in patients with AD is 90–95%, while the positive predictive value is less than 50%, suggesting that these tests are of high value in excluding food allergy [71,72]. Moreover, atopy patch testing (APT) to food allergens has been studied to evaluate non immediate eczematous exacerbations in children with AD. Although promising, they are not standardized yet and are not suggested in routine diagnosis of food allergy in patients with AD [73,74]. More recently, specific IgE levels and the pattern of reactivity to food components have been proposed to distinguish AD food allergic from tolerant subjects. Seventy-eight children with moderate-to-severe AD were assessed due to a history of clinical reactivity to milk, egg, peanut, wheat, and soy, showing that 91% percent of them were sensitized, while 51% reported allergy to at least one of these five food allergens. The IgE levels corresponding to each of these foods as well as the levels of specific components (Bos d8: milk casein, major allergen in cow’s milk; Gal d1: ovomucoid major allergen in Hen’s egg; and Ara h2: 2S albumin, major peanut allergen) were significantly higher in the moderate-to-severe AD allergic group [75].

### 6.3. The Role of Elimination and Reintroduction of Certain Foods in Diagnosis of Food Allergy

In case of exacerbation of AD due to a potential food allergen, in the absence of an IgE-mediated reaction, an elimination diet is still recommended by the European Academy of Dermatology and Venereology recent guidelines [27]. Due to a potential placebo effect, which may accompany the elimination, a reintroduction after a short period of time in which an improvement is observed is suggested in order to confirm the diagnosis, and then, a strict elimination diet could follow as long as skin moisturizing is ensured as the first step of standard of care [66].

## 7. Elimination Diets in AD: Favor or Harm?

### 7.1. Elimination Diets May Have a Beneficial Effect on AD Severity

Elimination diets based on positive oral food challenges have been suggested to reduce the severity of reluctant atopic dermatitis’ symptoms. This strategy can be implemented in the case that a specific food is a possible chronic trigger or in the case of a confirmed IgE-mediated food allergy [76]. Respective data indicate 60–80% improvement in those patients, as assessed by the skin surface area affected, sleeplessness, antihistamine use, and pruritus [77,78].

### 7.2. Elimination Diets May Increase the Risk of Food Allergy in Patients with AD

There is ample evidence that the removal of a previously consumed food in patients with AD can result in IgE-mediated food allergy upon reintroduction, as sensitized subjects can lose immune tolerance [79]. Removal of milk allergen in order to improve AD in children who previously tolerated milk for a median of 2.3 years resulted in occurrence of immediate reactions in double-blind placebo-controlled food challenges upon reintroduction in all children [80]. Fatal IgE-mediated reaction following reintroduction after a long-term elimination diet due to persistent eczema was reported in an eighteen-year-old girl who was sensitized although previously tolerant to milk proteins [81]. Trying to characterize patients who developed food allergy to previously tolerated foods, Nachshon et al. reported that the onset of food allergy to previously well-tolerated foods is more frequent among patients with severe AD [82]. Furthermore, elimination diets, especially in patients with multiple sensitizations to foods, have been associated with nutritional risks, subsequent growth failure, and nutritional deficiencies [83], while a substantial impact on patients’ and their families’ health-related quality of life also exists [38,84,85,86].

Conclusively, elimination diets should be tailored not only to improve AD severity but also to avoid negative impacts ranging from nutritional risks to life-threatening allergic reactions. Thus, a well-balanced decision should be made in the case that a food is clearly identified as an AD exacerbation trigger and after informing the patient and the family for the limited yet existing benefits and the potential harms of an elimination diet.

## 8. Aiming to Protect: Is There a Way to Interfere in the Relationship between Food Allergy and Atopic Dermatitis?

Regarding preventive strategies for food allergy and AD, studies have focused on early restoration of skin barrier impairment and/or early introduction of food allergens in high-risk children to promote food tolerance [86,87,88,89,90].

### 8.1. Restoring the Skin Barrier and Preventing Atopic Dermatitis Could Reduce Epicutaneous Sensitization and Subsequent Food Allergy

Pilot studies have shown that paraffin-based emollients can reduce the risk of developing AD up to 50% [91,92]. However, two prospective open-label, interventional, randomized controlled trials (RCT) failed to prove the potential protective effect of early emollients use in AD development. The BEEP (Barrier Enhancement for Eczema Prevention) and the PreventADALL (Prevent Atopic Dermatitis and ALLergies) studies evaluated whether the application of petrolatum-based moisturizers or bath oils from the first weeks of life could prevent AD [93,94]. Both studies failed to show a significant reduction in AD rates; in the PreventADALL study, neither early skin emollients nor early complementary feeding reduced the development of AD by the age of 12 months, while in the BEEP study, there was an increased rate of skin infections. Moreover, the use of emollients and symbiotics alone or in combination were insufficient to prevent the occurrence of AD and food allergy at the age of 12 months [95]. These unexpected results of RCT raised numerous questions and highlighted once again the complex nature of AD and food allergy. The limitations of these studies, including low adherence rates, late age of first application, and nature of the emollients as well as lack of targeting the inflammation part of AD, may partially explain the contrasting to pilot studies’ results.

Inconsistently, preliminary findings from PEBBLES (Prevention of Eczema By a Barrier Lipid Equilibrium Strategy), a pilot randomized, parallel, single-blind controlled study that used a ceramide-containing cream in high-risk infants from birth up to 6 months, showed a reduction in AD incidence at 12 months and as well as in food sensitization at 6 and 12 months in the treatment group compared to controls [87]. A RCT is ongoing (PACI trial) to determine whether early aggressive treatment of AD with a combination of emollients and topical corticosteroids in infants may prevent the development of subsequent allergen sensitization and associated allergic disorders such as food allergy, asthma, and allergic rhinitis [88].

Despite controversial results of existing RCTs [93,94], recent promising preliminary results support the hypothesis that early aggressive treatment of AD in high-risk infants may not only prevent AD development but also reduce rates of food sensitization and food allergy later in life [87,88].

### 8.2. Early Introduction of Allergenic Foods in High-Risk Infants to Prevent Food Allergy and Atopic Dermatitis

The landmark LEAP (Learning Early about Peanut Allergy) study was the first randomized controlled trial (RCT) aiming to evaluate the avoidance versus early introduction of peanut consumption in high-risk infants (4–11 months) with moderate-to-severe AD and/or egg allergy. When assessed with double-blind placebo-controlled food challenges at the age of 5 years, toddlers in the consumption group had significantly lower incidence of peanut allergy compared to the avoidance group (1.9% vs. 13.7%), with an overall mean reduction of peanut allergy by 81%. This study was the first interventional study to clearly state that food allergy could be prevented in high-risk children—such as those with moderate-to-severe AD—with allergic potential to develop food allergy. Exposing the infants to food allergens early through the intestinal immune system rather than the inflamed skin could reduce food allergy risk [86]. A follow-up study of the same cohort (LEAP-On) showed that in children who consumed peanuts until 60 months of age, a subsequent 12-month avoidance did not increase the risk of peanut allergy, suggesting that such a strategy might have a long-term positive effect [96,97]. It should be noted, however, that early peanut introduction had no impact on AD trajectory over time. Moreover, introduction of potential food allergens in high-risk infants should be performed following an allergy work-up, including food-specific IgE measurement and/or skin prick tests, due to the risk of IgE-mediated reaction in already-sensitized infants with severe AD [70,98].

Conclusively, the early introduction of food allergens and especially peanuts has been shown to reduce food allergy rates attributed to specific food allergens, but no effect was shown in the AD course over time. Thus, the strategy of early introduction of allergenic foods in infants’ diets may have a beneficial effect in reducing food allergy incidence, but possible effects to AD are yet to be discovered through well-designed RCTs.

## 9. Conclusions

There is a complex interplay between food allergy and atopic dermatitis, with growing evidence supporting that atopic dermatitis plays a role in the food allergy causality pathway and not vice versa. Atopic dermatitis is a chronic inflammatory skin disorder, and food allergy is only one of its potential triggers. Skin care should always be a priority when assessing a difficult to treat eczema. In case an allergic work up is required, it should be done carefully and in a targeted manner, as asymptomatic sensitization is quite common. The role of oral food challenges is once again the gold standard in diagnosing food allergy, and physicians should have in mind that in selected cases of AD, it is necessary to assess the patients for food allergy, especially when the immediate-type allergy is not supported by the clinical assessment. Although in the forefront for decades, elimination diets are now believed to carry limited benefits and potential harms in management of AD. Therefore, a well-balanced decision when it comes to avoidance of a potential allergen to control AD should be made. Physicians should always be aware that in order to improve AD, the avoidance of a previously tolerated food can turn into a potentially life-threatening allergen. Hence, elimination diets carry more risks than benefits if not used in well-selected AD cases.

## Figures and Tables

**Figure 1 jcm-11-04232-f001:**
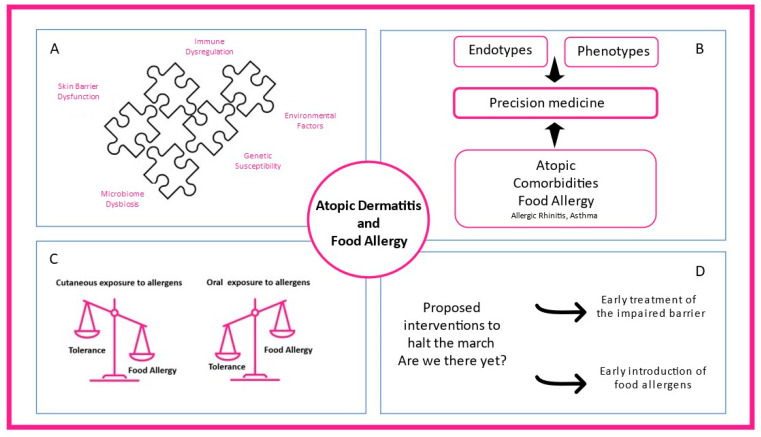
The complex interplay between atopic dermatitis and food allergy. (**A**) Skin barrier dysfunction, microbiome dysbiosis, immune dysregulation, environmental factors, and genetic susceptibility contribute to atopic dermatitis heterogeneity. (**B**) Different endotypes, phenotypes, and atopic comorbidities present in clinical practice shaping precision medicine interventions. (**C**) While oral exposure to food allergens through the gastrointestinal tract promotes food tolerance, cutaneous allergen exposure can lead to food allergy. (**D**) Early treatment of the impaired skin barrier and early introduction of food-specific food allergens in high-risk infants are proposed mechanisms to prevent atopic dermatitis and food allergy.

**Figure 2 jcm-11-04232-f002:**
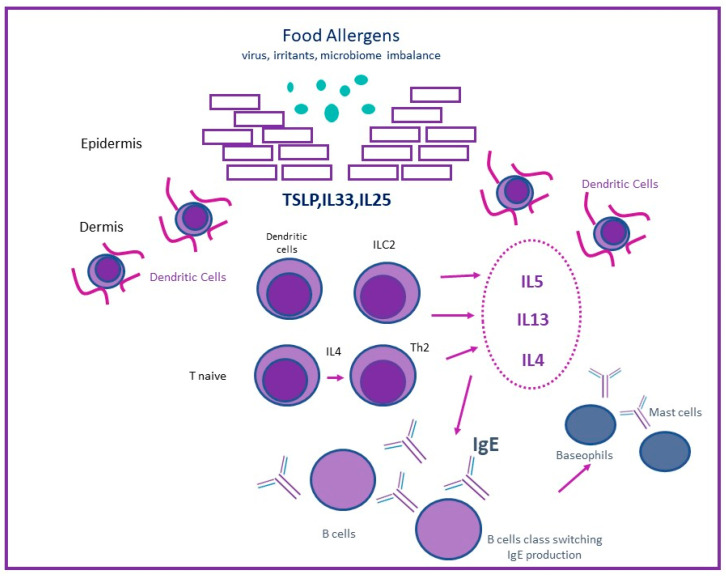
Penetration of food allergens through the impaired skin. Impaired skin barrier in AD leads to skin inflammation and clinical atopic dermatitis. Exposure of food allergens through the impaired inflamed skin leads to penetration through skin, sensitization, and food allergy.

**Figure 3 jcm-11-04232-f003:**
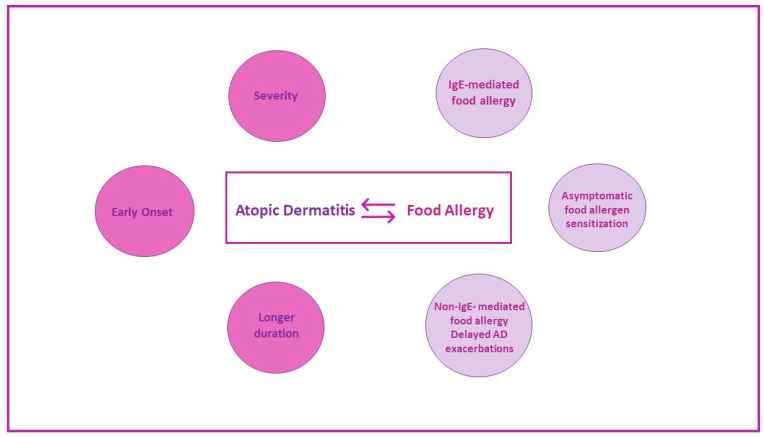
Different patterns of interaction between atopic dermatitis and food allergy. A close relationship between atopic dermatitis (AD) and food allergy, especially in childhood, is well-established. Infants with severe AD, early-onset AD, and a longer duration of AD have a much higher risk of food allergy. Food allergens’ penetration through the impaired skin can lead to asymptomatic sensitization, clinically relevant sensitization and food allergy, or delayed exacerbation of AD, which is thought to be non-IgE-mediated.

## Data Availability

Not applicable.

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
