# Peer review of "Atopic Dermatitis and Food Allergy: A Complex Interplay What We Know and What We Would Like to Learn"

_jcm, 2022, doi:10.3390/jcm11144232_

Round 1

Author Response

''Please see  the attachment''

Reviewer 2 Report

do not use abbreviation such as FA. food allergy is not long. FA give confusion.

do not use your own abbreviation like DBPCFC and ... 

It can be important review. However, it is difficult to follow the author's intention. It will be better if author provide subtitle in each section and short conclusion. 

1.2. Phenotypes and endotypes of AD

how phenotype and endotypes are related with food allergy. Has “six cluster classification” any meaning with food allergy?  If there are not big relationship, shorten this portion. 

2. Association between food sensitization and food allergy with atopic dermatitis

           What is conclusion? 

 3. Mechanisms explaining how atopic dermatitis promotes sensitization to food allergens 

The dual allergen exposure hypothesis 

It can be divided into several sections for easy understanding.

3.1. introduction

3.2. exposure through skin

3.3. exposure through oral mucosa

   if 3.2. and 3.3. is well compared, it can increase the readers’ understanding.

May “skin microbiome story” go to exposure though skin? 

3.4. conclusion 

4. Patterns of clinical reactivity to foods in children with AD 

Although 3 patterns are introduced, explanation about 3 patterns are not enough. 

provide subtitle. 

5. Diagnosis and management of food sensitization and food allergy in patients with AD

It can be much easy to understand if contens are divided according to its main message. In other words, it is not easy to connect the main message of each paragraph. 

provide subtitle as possible as you can.  

6. Elimination diets in AD: favor or harm?

provide subtitle and short conclusion. 

7.1. Restoring the skin barrier and prevent AD could reduce epicutaneous sensitization and subsequent food allergy

So, what is short conclusion?

7.2. Early introduction of allergenic foods in high-risk infants to prevent FA 

So, what is short conclusion?

8. Conclusion

Confusing conclusion. 

It should be clear and simple. 

Author Response

''Please see the attachment''

Round 2

Author Response

Please See the attachement. 
